# Past Human Mobility Corridors and Least-Cost Path Models South of General Carrera Lake, Central West Patagonia (46° S, South America)

Paulo Moreno-Meynard, César Méndez, Iñigo Irarrázaval and Amalia Nuevo-Delaunay *

Centro de Investigación en Ecosistemas de la Patagonia (CIEP), José de Moraleda 16,
Coyhaique 5951369, Aisén, Chile
* Correspondence: amalia.nuevo@ciep.cl; Tel.: +56-67-2247808

**Abstract:** Understanding the use of natural corridors is critical for characterizing the past use of marginally occupied landscapes at the Andean fringes of western Patagonia by the hunter-gatherer groups who inhabited this region. In this paper, we combine least-cost path models and archaeological surveys and excavations to determine the possible movements along the southern margin of General Carrera Lake. The methodology includes defining uncertainties that allow for modeling a set of equiprobable routes, thereby avoiding problems with errors and biases from predictors, such as slope, land-use cover, and seasonality. The results identify mobility corridor bottlenecks, i.e., geographical areas with a high probability of travel routes with equiprobable routes that converge. In addition, we identify areas where travel routes are likely to diverge into multiple semi-parallel routes. The study of archaeological sites provides stratigraphic data to control for the chronology and characteristics of mobility along this transect. The correlation between archaeological sites and paths, specifically for the control and winter scenarios, shows the quality of these route predictions. These results indicate the repeated use of internodal spaces that were effectively incorporated into mobility during the last three millennia.

**Keywords:** past mobility; cost surfaces; stochastic variability; hunter-gatherer routes; late Holocene; Central West Patagonia; least-cost paths

## 1. Introduction

Central West Patagonia is a region of notable landscape and climatic contrasts lying on the fringes of the southern cone of South America. Human groups, namely, hunter-gatherer societies, made use of these spaces by accessing them from the eastern steppes, most likely through seasonal movements throughout the Holocene and as a result of population growth [1]. However, this use was not uniform or continuous, as was expected perhaps in more open spaces, but rather organized through natural corridors based on prominent landscape features and as indicated by the distribution of the archaeological record [2,3]. The hunter-gatherer movement does not occur randomly across space but rather is organized along routes based on their acquired knowledge of the landscape [4–6]. The archaeological study of movement along routes or beyond the nodes of more permanent use has grown increasingly over the last few years [7]. The use of internodal spaces by hunter-gatherers has been studied in Patagonia by combining site distribution, provenance of exotic goods, rock art and stable isotope information, and by considering the role of biogeographic constraints for mobility [8]. Rock art, a paramount feature in Patagonia and in our study area, has been shown to occur along mobility corridors using geographic information systems, supporting the idea that these were associated with trails [9]. This paper moves beyond the distribution of sites as indicators of the location of human signatures to considering geographical attributes of a given landscape for justifying the selection of specific routes

actively used by past societies. We use multiple simulations of least-cost path (LCP) routes to present plausible mobility corridors along the southern coast of General Carrera Lake.

LCP models aim to find a route between two points minimizing variables such as effort, time, and/or distance. Commonly, LCP models optimize the travel cost as a function of terrain slope (where steeper is more costly) and the relative feasibility of movement through specific land cover types (e.g., dense forest, shrubs, bare soil, snow, etc.). LCP analysis has provided valuable insights into the understanding of past travel routes [10–14]. Some applications have been used in understanding the mobility and landscape choices of hunter-gatherers with results that allow predicting new findings [15]. However, there are still many LCP model parameters and assumptions that can influence the travel route [12]. For example, LCP analyses are based on today's concepts of travel optimization, and travel routes may differ depending on factors such as load (cargo), weather conditions, and the quality of datasets [16]. LCP models are considered as a stage in the understanding of movement and needs to be associated with other sources of evidence for building a comprehensive view of past mobility.

Another source of uncertainty comes from digital elevation model resolution and errors [13]. For example, large-scale digital elevation models, such as SRTM and ALOS, are available at 30 m and 12.5 m spatial resolutions (pixel sizes) [17,18]. Topographic features such as creeks and gullies are not adequately represented at this scale. Therefore, models that output a unique route could be biased toward certain mobility cost criterion or model assumptions, and interpretation is essential to account for such assumptions on past travel routes [12,13,19].

Different strategies have been used to address some of the main limitations of LCP models. Here we highlight two: circuit theory [20,21] and running multiple LCP realizations [19]. The first approach, circuit theory, aims to identify areas of high/low mobility potential (high/low current) across all landscapes. The output is not limited to a single path but a surface (raster) indicating potential mobility for each pixel. The tool helps explore landscape corridors. However, one of the limitations on human mobility applications is that circuit theory does not consider the overall connectivity between two specific locations. Therefore, areas of high mobility potential may be isolated or far from the optimal travel routes and not considered for travel [21].

The second approach, multiple LCP models, is based on running several LCPs where the input parameters are modified for each simulation. For example, Lewis [19] simulated the uncertainties of digital elevation models to obtain multiple realizations for the study area. Then, each digital elevation model was used to compute a LCP obtaining an ensemble of LCPs that represent a mobility corridor. Even though circuit theory and multiple LCP methodologies differ, both approaches aim to find multiple paths or geographical areas with high passage probability and less dense passages areas. Furthermore, both approaches are limited on assumptions such as the definition of mobility criterion and model parameters.

Different applications of these methods have been used for addressing mobility at different scales and both in the Atlantic slope, as well as on the western areas of southern Patagonia [22,23]. The human occupation of Central West Patagonia dates to the Pleistocene-Holocene transition (12,000 to 11,000 cal BP) and has strong ties with earlier populated demographic nuclei of the eastern extra Andean steppes, nowadays Argentine territory [24,25]. As such, the occupation of fluvial valleys and other Andean landscapes to the west has been interpreted as corridors that were articulated for the seasonal mobility of hunter-gatherer bands residing more permanently in the east [3,26]. The use of such corridors is a key feature for understanding how geographically marginal spaces were incorporated into the comprehensive picture of human occupation in the macroregion [1]. The archaeological record along these corridors is characterized by an east-to-west diminishment of artifact deposition, as well as occupations limited within specific time periods [3,27–29]. However, most characterizations of these corridors remain untested using methods derived from geography, particularly LCP models.

Ethnographic and ethnohistorical information supports a more permanent use of eastern extra Andean landscapes by indigenous hunter-gatherer societies that inhabited continental Patagonia [30,31]. Permanent dwellings and most of the movement occurred across the steppe as is reflected by territories redundantly occupied until the 20th century [32–34]. Mountainous and forested spaces of the west were regarded negatively and often avoided, despite the fact that archaeological records occur in these areas [35]. This is in agreement with the lack of systematic observations of indigenous populations during historical times in Central West Patagonia [36,37]. The occupation of this area was certainly less intense than neighbor steppes and likely seasonally restricted as has been described archaeologically [38]. In that line, the fact that there is no local or regional knowledge of indigenous inhabitants of Central West Patagonia makes us turn to models based on current spatial data for addressing past movement.

The main objective of this study is to determine the possible routes of movement along the southern margin of General Carrera Lake from east to the west through the identification of archaeological finds and LCP models. Earlier applications in Patagonia have been undertaken at a regional perspective using broad-scale datasets [22]. However, our basin-scale perspective seeks to understand fine grain mobility by selecting a study area where human movement is naturally channeled. The specific objectives are: (1) to present the archaeological record obtained through systematic surveys and excavations and (2) to generate mobility corridors from LCP models, including different variables and factors, such as slope, land use, and seasonality. The ultimate goal is to better understand the possibilities and constraints under which human mobility most likely occurred.

## 2. Materials and Methods

### 2.1. Study Area

Central West Patagonia is defined here as those basins that discharge into fjords or channels of the Pacific Ocean between 44° and 49° south latitude (Figure 1a). The climate of this region is mainly controlled by the westerly winds belt and its interaction with the Andean Cordillera [39]. On the one hand, the mountain range to the west generates an orographic effect, producing a marked west to the east reduction in rainfall, from more than 6000/4000 mm to 300 mm close to the Chilean-Argentinean border (Figure 1b) [40]. On the other hand, while temperatures are markedly lower at the two ice field glaciers over the Andes, low altitude valleys and lake shores present higher mean temperatures (Figure 1c). The study area corresponds to the southern margin of General Carrera Lake (Figure 1d) and shows a prominent vegetation gradient from temperate forests to the west to a Mediterranean steppe to the east [41]. According to the Köppen-Geiger climate classification, the area entails mainly temperate without dry season (Cfsk') and cold steppe (BSk'c) climates, whereas boreal without dry season (Dfk'c), frost (EF), and tundra (ET) climates are less represented [42]. Regional paleoclimate reconstructions indicate that after the end of the last glacial period (~11,600 cal BP), cooler conditions and variable precipitation prevailed until 8200 years ago, when precipitation increased to maximum values, allowing the expansion of forests in the region [43,44]. These conditions persisted until the late Holocene, which are characterized by a gradual opening of the forest, although under high interregional variability [43,45,46].

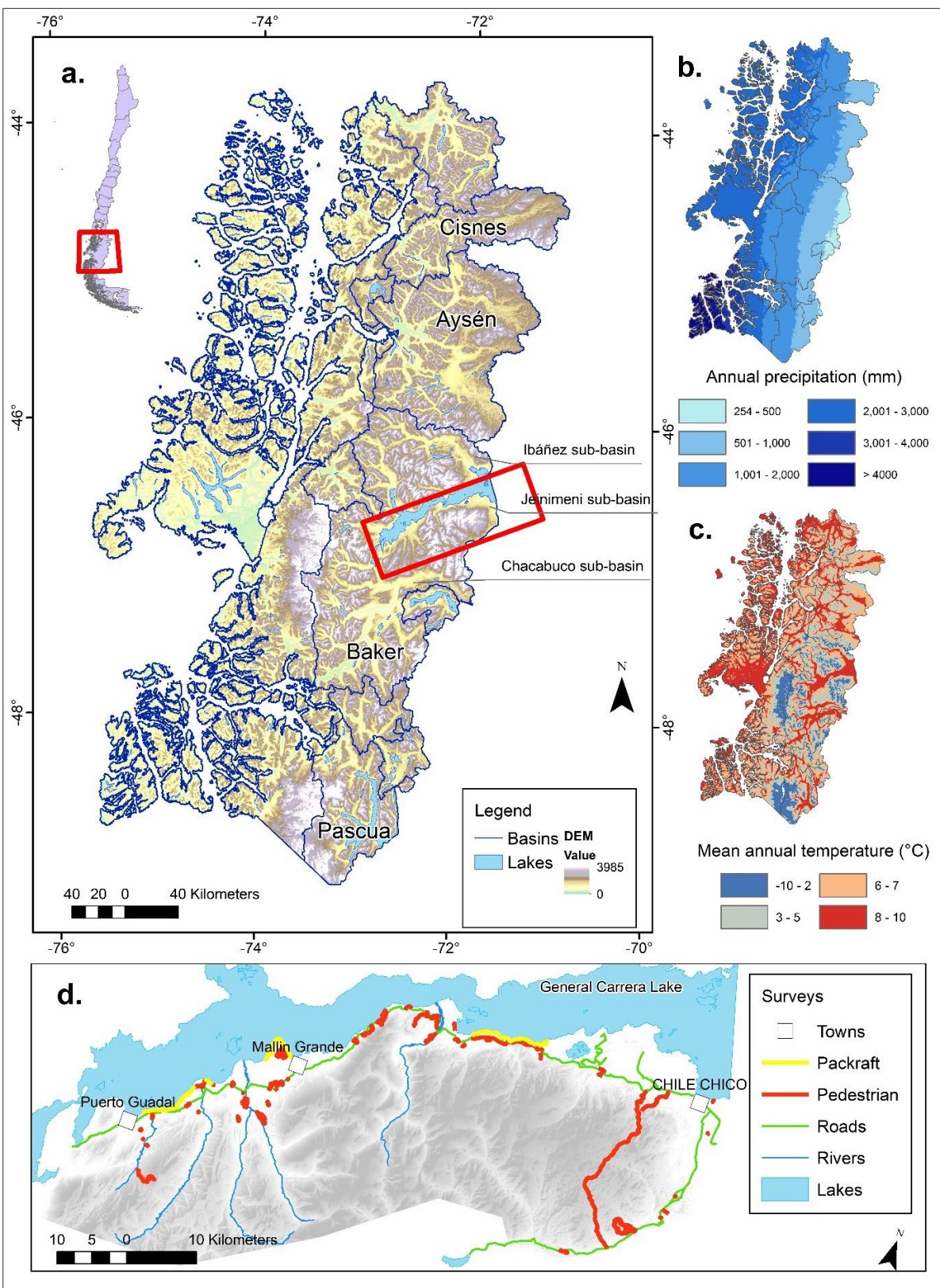

**Figure 1.** Map of the study area: (**a**) Central West Patagonia with information on its main basins, the small red box indicates region (a), and the red rectangle indicates study area (d); (**b**) raster representation of annual precipitation; (**c**) raster representation of mean annual temperature; (**d**) study area with the distribution of archaeological surveys.

### 2.2. Archaeological Material and Methods

Archaeological points of interest recorded in the area close to Chile Chico town total 178. These include mainly open-air concentrations of archaeological material, both in the surface as well as in stratigraphy, rock shelters, with and without rock art, funerary structures, and isolated surface findings [27,47–49]. These are spatially organized along two distinctive landscape transects: the southern coast of General Carrera Lake (*n* = 96; 53.9%), running northeast to southwest, and the Jeinemeni River (*n* = 58; 32.6%), running parallel to the international border, from south to north. These add to a series of surface findings (mainly lithic material) on top of the Jeinemeni Plateau (*n* = 24; 13.5%). The chronology for the occupation of this area extends from ~3550 to 370 cal BP as indicated by available dates for the southern coast of General Carrera Lake and the vicinity [27]. This subset was selected for analysis because it is bounded by the lake to the north and a steep topography to the south, thereby allowing assessing movement along a conspicuous east-to-west corridor. It also accounts for more than half of the sites of the area and features records distributed along all its extension. Table 1 summarizes the archaeological record of the southern coast of General Carrera Lake (the detailed location, altitude, and characteristics of each record are presented in Table S1).

**Table 1.** The archaeological record of the southern coast of General Carrera Lake.

| Site Type | *n* | Percentage |
| --- | --- | --- |
| Rock shelters | 49 | 51.04% |
| Open-air sites | 30 | 31.25% |
| Funerary structures | 9 | 9.38% |
| Isolated surface findings | 5 | 5.21% |
| Structures | 1 | 1.04% |
| Other | 2 | 2.08% |
| Total | 96 | 100% |

The archaeological record along the southern fringe of General Carrera Lake was assessed through pedestrian (318.8 km) and by portable inflatable boat, also known packraft (42.5 km) and surveys conducted along the mountain slopes and the coast of the lake, respectively. The surveys were carried out between the towns of Chile Chico (46°32′39.33″ S; 71°41′59.69″ W) and Puerto Guadal (46°50′49.28″ S; 72°42′10.91″ W), which are separated by 82.6 linear km (Figure 1d). Surveys were intensive, as they considered investigating almost every option for archaeological records and potential paths. They benefited from the knowledge of local people and were extended for three years. In addition to pedestrian movement, they even incorporated the use of packrafts for accessing those areas along the lake shore where land movement was restricted, and archaeological findings were negative. Moreover, there is no indication of navigation by past inhabitants of the region; therefore, an archaeological record in places not accessible by pedestrian movement is not expected.

Surveys included recording isolated surface finds and larger archaeological concentrations with standardized methodologies used throughout the region [50]. These methods include surveying areas at comparable intensities, utilizing transects within this areas, geo-localizing each finding, and recording qualitative and quantitative attributes at each site [51]. Collection included sampling for toolstone and artifact class diversity, bone material for taphonomic characterizations, and other less-represented material types. Rock art sites are crucial features in this area, comprising most of the evidence gathered. Their frequency, density, and formal characteristics were recorded by using standardized field parameters [52]. One such site is the La Tina (RJ82) rock shelter, which was selected for excavations (2 m$^2$) given its strategic location along the surveyed transect. The excavation was performed following stratigraphic units defined through contact surfaces with one another and changes in sedimentation. The process included the three-dimensional piece-plotting of all features, with all evidence larger than 2 cm, and each radiocarbon dating sample [25].

Radiocarbon AMS dates were processed in the Direct AMS Laboratory. All results were corrected for isotopic fractionation with an unreported $\delta^{13}C$ value measurement of the carbon prepared by the accelerator. All dates were calibrated at 2σ with the Calib 8.1 program applying the ShCal20 curve and are expressed in calibrated years before the present (cal BP) [53,54].

*2.3. Least-Cost Path Model*

The methodology to derive the mobility corridors is based on LCP models. First, we define the mobility cost criterion to travel along the landscape, namely the cost surface. Here we define three cost surface scenarios. Second, we derived the LCP following a deterministic procedure, i.e., for each cost surface scenario, and one optimal route is obtained, as is commonly done in LCP models. Finally, for each LCP simulation, we incorporate a random or stochastic term representing the uncertainties of the mobility cost criterion. The stochastic term adds variability to the outputted LCP, resulting in a set of equiprobable routes for each scenario.

2.3.1. Cost Surface Scenarios

LCP models aim to find the optimal route between two points across a landscape. The first key step is to define a cost surface. The cost surface represents the degree of human pedestrian effort (represented through various measures, such as time or calories) involved in moving across the landscape and is generally defined as a function of slope (e.g., steeper slopes require more effort) and land use (e.g., snow requires more effort than a bare surface).

The hunter-gatherer movement in western Patagonia was organized along corridors; many of them are west-to-east landscape features that break the mountainous terrain [2,23]. Given the greater antiquity and higher concentration of human occupations to the east, these corridors have been interpreted as having been traversed by hunter-gatherers from nodes located at the eastern extra Andean steppes. A key point to build cost surfaces is to start with a simple cost surface model and to refine it in a consecutive procedure [12]. Moreover, some attributes, such as snow presence and altitude, are described as necessary variables to incorporate into a cost surface [12]. To derive mobility corridors from the Jeinimeni River outlet (east of Chile Chico town) to the westernmost limit of General Carrera Lake (herein Puerto Guadal) in a northeast to southwest transect, we consider three cost surface scenarios, building the mobility cost criterion by adding variables or factors with different weights:

- Scenario 1 (control) uses slope as the only criteria to build the cost surface and for deciding a possible path. Scenario 1 is conceptualized as the standard scenario.
- Scenario 2 (summer) represents the summer season, and the cost surface is defined by a combination of slope (70%) and land use (30%).
- Scenario 3 (winter) represents a model of the winter season mobility corridor by incorporating snow coverage as an additional restriction for human movement.

The cost surfaces are derived as follows. First, a slope map is computed based on the ALOS 12.5 m digital elevation model [18] and scaled into a slope cost surface, where steeper slopes are more costly than flat terrain. Then, slopes are categorized into nine classes where lower slopes have lower values (i.e., 0–5° (1), 5–15° (2), 15–25° (3), 25–35° (4), 35–45° (5), 45–55° (6), 55–65° (7), 65–75° (8), and 75–85° (9)). Second, water bodies such as lakes are defined as implausible options for movement. Third, a land-use cost surface was obtained from the land cover map of the Chilean National Land Use [55], and we assigned a relative mobility cost value to each land cover class. The use of a current land use coverage map is the best approximation to the historic land use, even with the inclusion of inexistent classes, such as, agricultural, or urban areas, that are minimally represented in the study area today. The relative travel cost was defined using expert knowledge assimilating the relative ranking from other studies [16,56] (Table 2) and considered criteria such as vegetation density (e.g., close vs. open forest) and soil condition (wetland vs. bare surface). The

Scenario 2 cost map was obtained by a weighted sum of 30% for the land-use cost map and 70% for the slope cost map. Last, for Scenario 3, we summarized 10 dimensionless units to each pixel over the tree line to simulate snow cover, thereby increasing the difficulty of movement across highlands.

**Table 2.** Relative travel cost for each land cover class.

| Landcover | Relative Travel Cost |
| --- | --- |
| Steppe | 1 |
| Agricultural, grasslands, urban areas, bare highlands | 2 |
| Open shrublands, no vegetation lands | 3 |
| Old-growth forests, arborescent shrublands, close shrublands, second-growth forests, tree plantation | 4 |
| Stunted forests | 5 |
| Landslides, riverbanks | 6 |
| Wetlands | 7 |
| Snow | 8 |
| Rocks | 9 |

### 2.3.2. Deterministic Least-Cost Paths

Once the cost surfaces are determined for each scenario, it is necessary to find an optimal route between two points that minimizes effort across the previously defined cost surface. An extensive tool used in archaeology to model those routes is the LCP algorithm [11–16]. We used the LCP algorithm from Spatial Analyst Tools of ESRI ArcGIS Desktop 10.8 that include the cost distance and cost path tools. These tools are based on Dijkstra's algorithm using the back-link predecessor, avoiding the effects of the steepest descent approach. In addition, they have an optimal internal model for dealing with anisotropic cost calculations [12]. The number of grid neighbors considered was eight, and we used the default software parameters from the cost distance tool.

### 2.3.3. Stochastic Least-Cost Paths

LCP models are dependent on mobility cost criterion (input data) which have uncertainties and biases [16,19]. Specifically, the ALOS digital elevation model incorporates an unknown error into our cost surfaces. On the other hand, the mobility cost criterion using land-use cover has errors, and the expert definition of relative travel cost for each land cover class has an undetermined bias.

To add variability in the cost surface, we added a spatially explicit random noise or stochastic field (Figure 2), and then we ran the LCP algorithm. The random noise was modeled as a spatially correlated Gaussian random field with exponential covariance. Similar geostatistical methods are often used for simulating uncertainties and/or unknown heterogeneity on surfaces or materials, flow routing algorithms, etc. [57,58]. It is noteworthy that no variability is incorporated if the noise term is too small. In contrast, if the added noise is too large, then the LCP is dependent primarily on the noise term. Therefore, we first needed to perform a sensitivity analysis to balance the noise magnitude. This was done through testing the model by gradually increasing the Gaussian random field variance (magnitude of the noise term) and testing different integral scales (spatial correlation) until enough variability was reached. Once the variance and integral scales was defined, the Gaussian random field was generated and added to the cost surface. For this step, we used MATLAB (R2019b) software; the results of the sensitivity test are presented in Appendix A. After obtaining the cost surface with random noise, the LCP was obtained as was explained in Section 2.3.2 [59] (Figure 2). Note that this least accumulative cost distance map from the cost distance tool is an intermediate product that informs about the accumulated distance from the start point toward the end point.

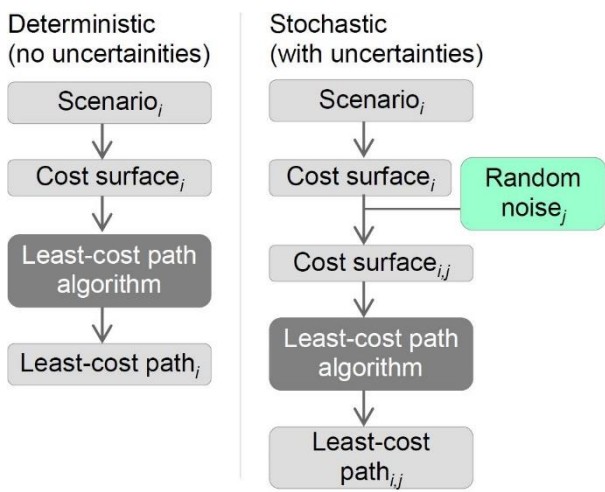

**Figure 2.** Workflow for deterministic and stochastic LCP models. Subindex *i* implies a different scenario. Subindex *j* identifies a specific random noise.

To summarize the LCP methodology, we first generated three cost surface scenarios. Then, we ran the following experiments based on those cost surfaces:

1.  Deterministic (without uncertainties). LCPs were obtained for Scenarios 1, 2, and 3 without adding uncertainties, and the output corresponds to one LCP for each scenario ($i \in (1, 2, 3)$ in Figure 2).
2.  Stochastic (with uncertainties or variability). The LCPs were obtained for Scenario 2 and Scenario 3 ($i \in (1, 2)$ in Figure 2). For each scenario, we performed one hundred simulations ($j \in (1, 2, \ldots, 100)$ in Figure 2), where for each simulation, a different *j* random noise was added to the *i* cost surface. Then, the output or mobility corridor corresponds to an ensemble of a hundred likely routes.

### 2.4. Validation of Deterministic Least-Cost Paths

There are several methodologies for validating potential routes, such as comparing them to historical or modern-day travel paths [12,19], understanding the location of critical points within a spatial context, and performing statistical assessments of the data. We chose to use a statistical approach to spatial analysis by defining the null hypothesis of no relationship between the mean orthogonal distance of archaeological sites and the LCP corridor for each scenario. The null hypothesis was created using a computer simulation ($n = 1000$ simulations) of a dot/pin map of an independent random process (IRP), also known as complete spatial randomness (CSR) [60]. We used the Monte Carlo procedure to simulate the locations of the same number of points that were observed in the archeological record for the area. In each simulation (IRP/CSR), we calculated the mean orthogonal distance to the LCP corridor, obtaining a sampling frequency distribution to contrast with the archaeological record observed mean orthogonal length. The advantage of the Monte Carlo procedure is that it allows us to avoid problems such as edge effects because we are using the same study area to compare simulations and observations.

### 3. Results

### 3.1. The Archaeological Record of the Southern Coast of General Carrera Lake

3.1.1. Site Distribution and Characteristics

The archaeological record along the southern coast of General Carrera Lake is markedly influenced by the topographical characteristics of the area. Due to the steep terrain, many archaeological findings are associated with the available rock shelters, generally facing the lake shore (north aspect). Such shelters, often small in terms of their available area for covered activities, occasionally host rock art and have little to no surface material (Figure 3). Stratified deposits in such shelters have been recorded in some excavations,

for instance, at Pampa de la Perra (RJ97), where the archaeological deposit spans the last millennium [27]. Open-air sites have been recorded in association with water bodies, either permanent, such as the General Carrera Lake itself, or seasonal small lakes, but only where suitable (low relief) surfaces were available for occupation. In this sense, large parts of the coast, where the terrain abruptly descends on the lake shore, have resulted in negative archaeological findings. The fact that General Carrera Lake derived from the former Chalenko glacial paleolake whose level has been dropping over the last 15,000 years, a time before human occupation, rules out the possibility of submerged archaeological records in shallow coastal areas [61]. Although preliminary, the available chronology for the southern coast of the lake starts at ~3550 cal BP and extends discontinuously until 370 cal BP [27]. The archaeological material recorded both at sheltered and open-air sites is characterized by high-quality siliceous lithic material (82%), of which several varieties are locally available, and exotic obsidian (6%) [62,63]. Bone material shows marked signs of weathering and is restricted to some sites where local conditions preclude its destruction (e.g., Laguna El Peligro, RJ81). Pottery is infrequent and has been recorded only at the Chile Chico 1 (RJ80) site with an associated chronology of 420 to 370 cal BP, which is in agreement with other regional occurrences for this technology [64,65]. The archaeological record also includes dispersed funerary features, which are small stone heap concentrations, known as "*chenques*" that have been vandalized over the years. Some human bone remains, curated in local collections, and have been radiocarbon dated between 620 and 540 cal BP, a chronology also in agreement with available ages for this funerary pattern [66,67].

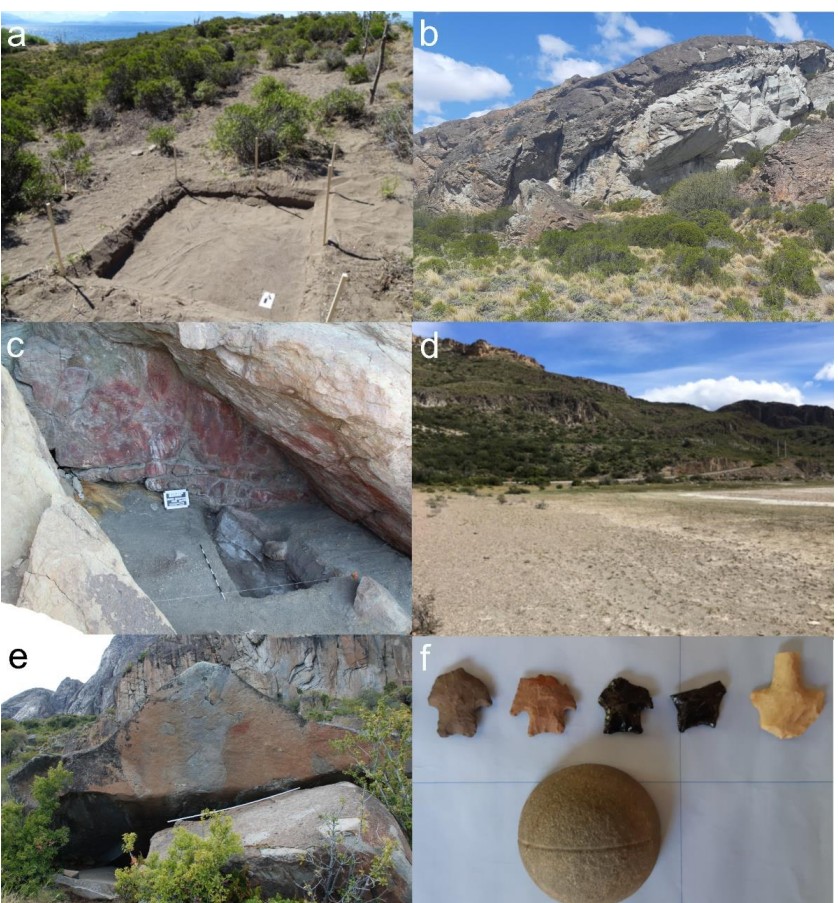

**Figure 3.** Site variability and the archaeological record along the southern coast of General Carrera Lake: (**a**) Chile Chico 1 site (RJ80); (**b**) Pampa La Perra (RJ97) site; (**c**) excavations at La Tina (RJ82) rock shelter; (**d**) Laguna El Peligro (RJ81) site on the border of the dry lake; (**e**) Guanaca y Luna site (RJ146); (**f**) characteristic lithic material from surface scatters, including fractured projectile points (silex and obsidian), a borer, and a stone hunting bola.

Rock art is one of the most conspicuous features of the archaeological record in this area. It has been recorded in the Jeinemeni River basin, where the Cueva de las Manos del Río Pedregoso site stands out as one of the largest concentrations of painting of the region [68]. However, along the southern coast of General Carrera Lake, rock art is observed as clusters of small sites with only a few motifs (e.g., at the Paso La Llaves sector). It consists mainly of red paintings, and motifs include geometric forms and animal depictions, the most prominent being the guanaco (*Lama guanicoe*) motif in locations such as the Guanaca y Luna site (RJ146). However, the dominant motifs are painted hands, with both positive (imprints) and negative (stencil) techniques. Among the recorded archaeological sites, La Tina stands out as the sole concentration of a high frequency of rock art, chiefly hand paintings.

### 3.1.2. La Tina Site Context and Chronology

The La Tina rock shelter was selected for excavations because (1) it has the highest rock art concentration on the southern coast of General Carrera Lake, (2) it is an intermediate point along the surveyed area, and (3) it has a potential excavation surface as opposed to other smaller or less sheltered sites where rock art was recorded. It is a large tuff promontory with steep topography and a limited sheltered area close to the rock wall (Figure 4a). Rock art occurs on high concentrations in six vertical walls (panels) and consists of a total of 84 motifs, among them at least 52 hand imprints made primarily with the stencil technique. Hand imprints sizes range from adults to infants. Only one schematic depiction of a guanaco was recorded, isolated from the rest of the hand representations. Almost all rock art at the La Tina site is red in color, except for a few cases of white and black motifs.

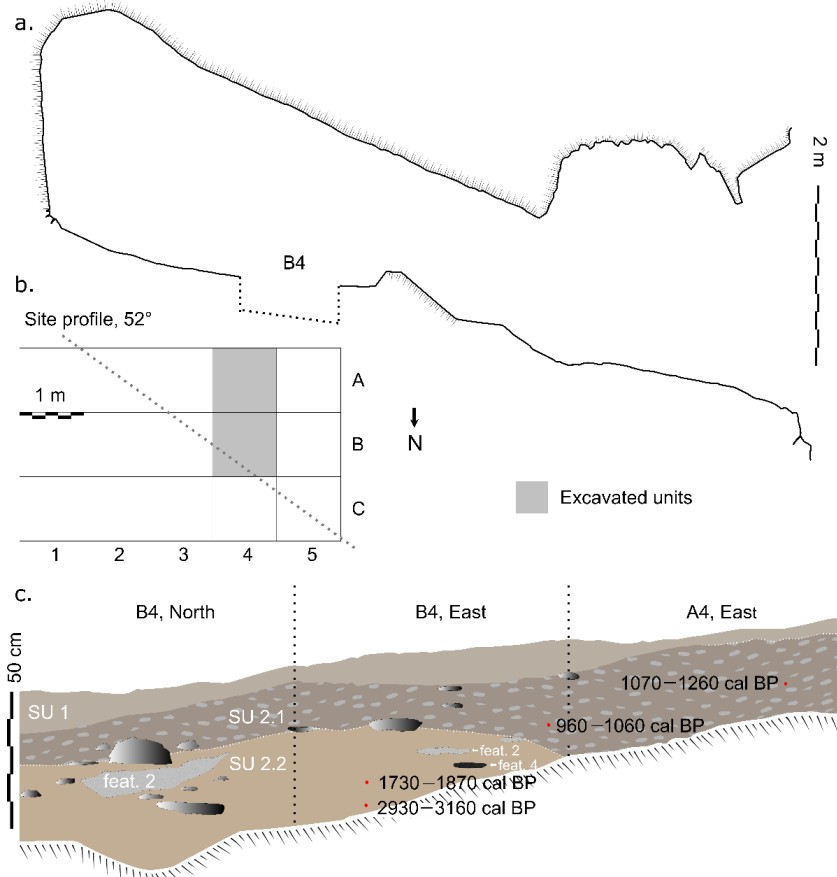

**Figure 4.** Archaeological features of the La Tina site; (**a**) cross section of the rock shelter; (**b**) excavated plan and location of the cross section of A; (**c**) stratigraphic section and radiocarbon chronology; feat: feature, SU: stratigraphic unit.

An excavation of 2 m$^2$ was located close to two of the main rock art panels to retrieve as much evidence from past occupations and potentially evidence of art production. The excavation revealed two main stratigraphic units (SUs) enveloping the archaeological record (Figure 4b). From top to bottom, SU1 is a friable, coarse, inorganic deposit of sand and gravel of eolian origin, likely volcanic, with few clasts (Figure 4c). SU2 is friable silts with varying contents of large to very large angular, matrix-supported clasts. Clast frequency, particularly low in the excavated base and to the east, allowed subdividing this unit into two subunits (SU2.1 and SU2.2). Although archaeological material was observed throughout SU2, most evidence was concentrated in SU2.2, including all observed features (hearths and ash concentrations). Material remains included lithics (mainly chipping debris), highly fragmented bone remains, small charcoal particles, very small clasts with traces of rock art, and pigment fragments. Four radiocarbon dates allowed us to confidently place the human occupations between 3050 and 1000 cal BP (Table 3), an age that is in agreement with other dates along the lake's coast and the neighbor Jeinemeni valley [27,69]. Although not considered to be the chronology of the manufacture of all paintings at the site, this age span is an appropriate fit with the expected time range for the production of the hand motif in the region and in wider Patagonia [70].

**Table 3.** Radiocarbon ($^{14}$C AMS) chronology for the La Tina rock shelter.

| Lab. Code | Unit | SU | Level | Age ($^{14}$C BP) | 2σ cal BP |
|-----------|------|-----|-------|-------------------|-----------|
| D-AMS 040162 | A4 | 2.1 | 15–20 cm | 1267 ± 22 | 1070–1260 |
| D-AMS 040163 | B4 | 2.1 | 15–20 cm | 1158 ± 21 | 960–1060 |
| D-AMS 040164 | B4 | 2.2 | 25–30 cm | 1908 ± 23 | 1730–1870 |
| D-AMS 040165 | B4 | 2.2 | 35–40 cm | 2931 ± 25 | 2930–3160 |

The sheltered area in La Tina is small and only suitable for short-term occupations [71]. The archaeological material in the assemblage is infrequent and does not represent the expectations for residential occupation or other activities. Features are also small, as expected for short stays, and the rock art is significantly abundant, indicating its production was likely the primary purpose for the human presence at the site. Occupations were of very low intensity throughout the ~2000-year span, which, considering the site context and the deposited assemblages, argues in favor of the site as an intermediate point along a mobility route. The fact that stenciled hand images dominate the rock art record suggests recognitions of specific persons who participated in the voyage.

### 3.2. Least-Cost Path Models
### 3.2.1. Deterministic Least-Cost Paths

The three scenarios described above in Section 2.3.1 showed a similar path at the beginning of the route (east), close to Chile Chico town, but with important differences after 72°10′ W. First, Scenario 1 (blue line in Figure 5), which incorporates only a cost surface derived from the slope, prefers spaces close to the lakeshore and avoids crossing major rivers, especially those with deep gorges (i.e., Avilés, Los Maitenes, and Las Dunas Rivers). Specifically, in the "Paso las Llaves", a major steepened sector located west of the Avilés River, the Scenario 1 route chose a very similar path to the current vehicular road that crosses this area. Second, the incorporation of land cover on the cost surface changed the route in many specific locations, preferring those land covers that are easier to transverse and avoiding sectors with rocks and dense vegetation. For example, the LCP for Scenario 2 (green line in Figure 5) chose to follow the Avilés River upward, thereby avoiding the steep and rocky "Paso Las Llaves". Similarly, to the west of Mallín Grande village, the route changed its trajectory with regard to the previous scenario by using land with less vegetation cover. Finally, the restriction of traversing through highlands during winter due to snow cover in Scenario 3 introduces a new detour that avoids going up the Avilés River and the lakeshore path by using a middle option in between both previous scenarios (red line in Figure 5).

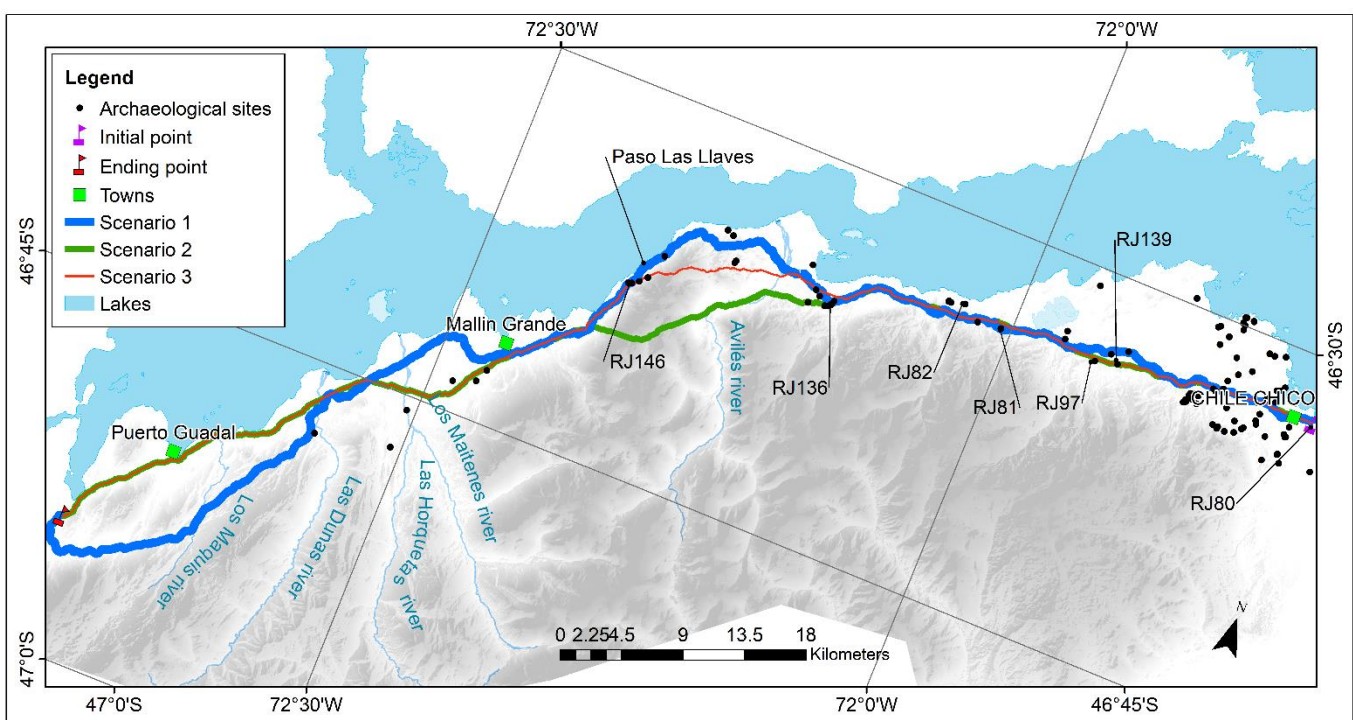

**Figure 5.** LCP routes for the three scenarios on the southern coast of General Carrera Lake. Archaeological sites: Chile Chico 1: RJ80; Laguna El Peligro: RJ81; La Tina rock shelter: RJ82; Pampa La Perra: RJ97; Casa Piedra: RJ136; *chenque* El Baño: RJ139; Guanaca y Luna: RJ146.

Monte Carlo testing through a computer simulation showed that two routes are located closer to archeological sites than IRP/CSR. The LCP derived from Scenario 1 is more spatially correlated with the archaeological record (*p* value = 0.0061), but the winter route (Scenario 3) also has a significant spatial correlation (*p* value = 0.0245). Thus, it is possible to associate the routes produced by both scenarios with past human paths or with mobility corridors. In contrast, the LCP derived from Scenario 2, associated with a highland route, does not present a spatial correlation with the assemblage of archaeological sites (*p* value = 0.1647). However, less sampling intensity at higher elevations cannot be ruled out as a potential source for bias.

3.2.2. Stochastic Least-Cost Paths for Summer and Winter Scenarios

The summer (Scenario 2) and winter (Scenario 3) mobility corridors were modeled by simulating 100 LCPs for each scenario. We describe the resulting mobility corridors considering the starting point in the east (Chile Chico town) and the end point in the west, separated by approximately 100 km. To describe the results, we divided the mobility corridor into three segments or clusters. First, from Chile Chico town (initial point) to the La Tina rock shelter corresponds to approximately 28 km. The second segment extends from the La Tina rock shelter to the Guanaca y Luna site at 25 km, and the last segment extends from the Guanaca y Luna site to the end point (47 km) (Figure 6a,d).

Starting from the east, the first 7 km of the summer mobility corridor present braided routes that span a 0.5–1.5 km wide corridor. Then, the corridor bifurcates into two branches (Point A in Figure 6c). The main branch (composed of 90% of the simulations) runs closer to the General Carrera Lake shore and consists of multiple braided routes spanning a 2 km-wide corridor. The main branch passes by archaeological sites *chenque* El Baño (RJ139) and La Tina rock shelter. The secondary branch runs semi-parallel to the main branch at approximately 300–500 m higher in elevation and as far as 1.5 km to the south from the main branch.

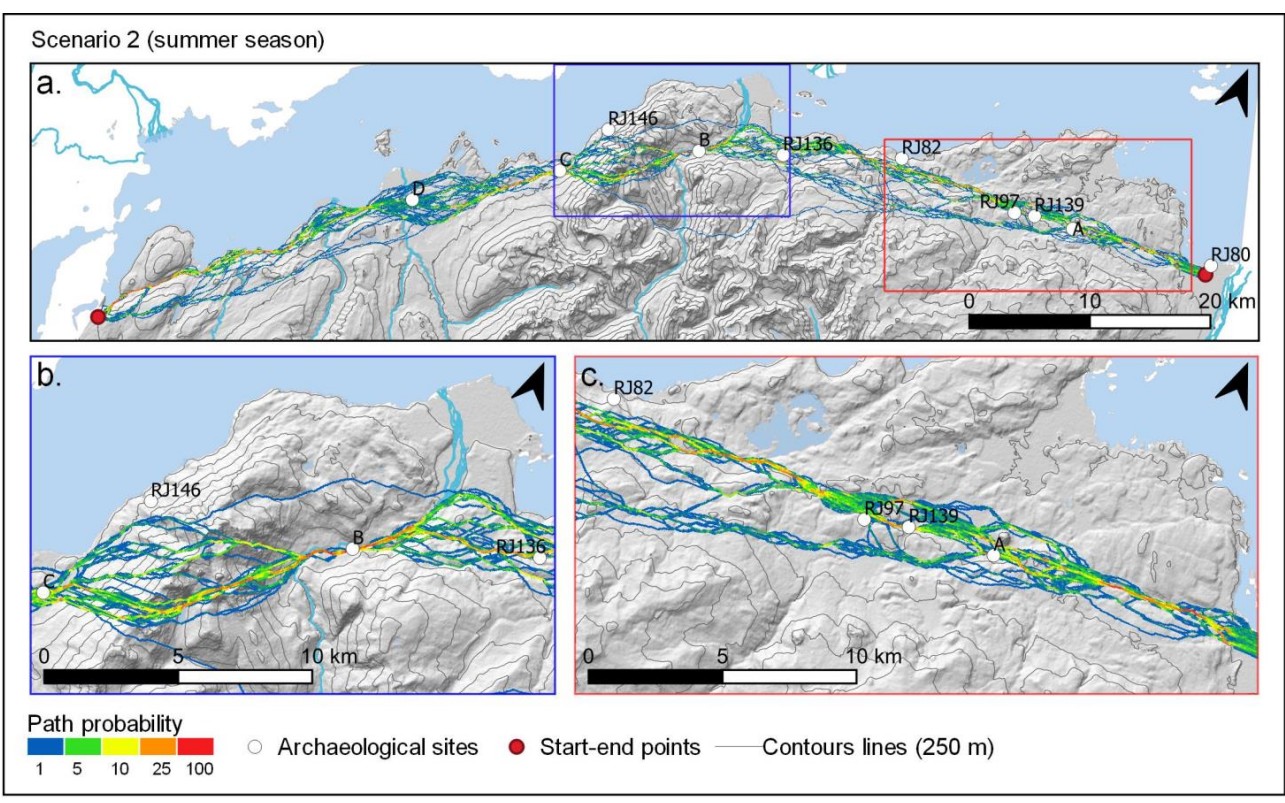

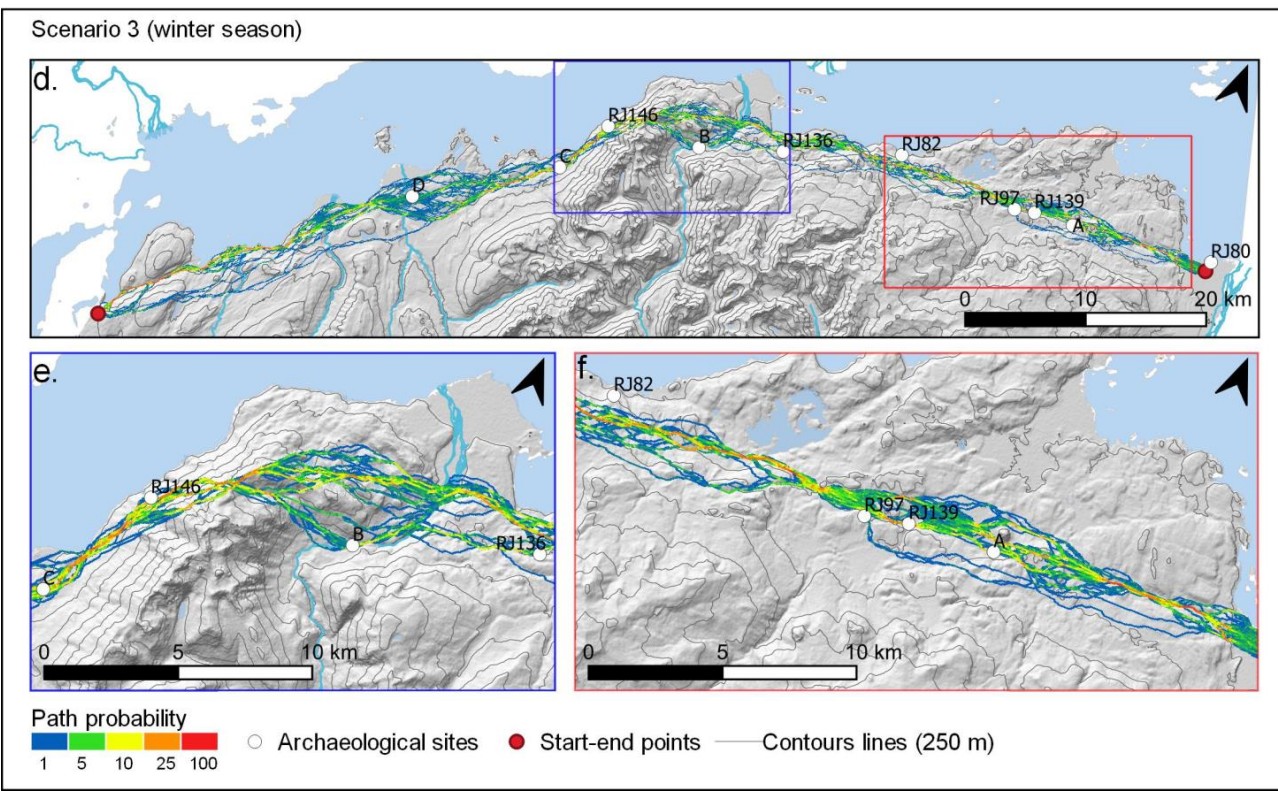

**Figure 6.** Summer (**a**) and winter (**d**) simulated corridors incorporating uncertainties in the cost surface. Specific clusters are presented: the Paso las Llaves sector (purple box: **b**,**e**) and Chile Chico town (initial point) to the La Tina rock shelter (red box: **c**,**f**). Archaeological sites: Chile Chico 1: RJ80; La Tina rock shelter: RJ82; Casa Piedra: RJ136; *chenque* El Baño: RJ139; Guanaca y Luna: RJ146.

After the La Tina rock shelter, both branches converge as the summer paths climb into the Avilés valley, and 98 of 100 simulations merge in a conspicuous bottleneck (Point B, Figure 6b). From the bottleneck, the corridor forks in multiple paths and traverses the mountainous terrain, reaching elevations above 1600 m a.s.l. Then, the corridor descends toward General Carrera Lake to converge in another bottleneck at elevations near 350 m a.s.l. (Point C in Figure 6b). In this segment, the corridor passes by the Casa Piedra (RJ136) site, and approximately 75% of the simulations pass more than 4 km south of the Guanaca y Luna site. Last, after Point C (Figure 6a), the corridor follows the shore of General Carrera Lake in braided routes, which diverge and spread when entering low relief areas (i.e., Mallín Grande, Point D in Figure 6a). Close to the end point, the pathways show one predominant branch running near the lake and a secondary branch (approximately 300 higher in elevation) before converging to the finish point.

The first kilometers of the winter scenario show braided pathways spreading in a 1–2 km-wide corridor until converging in a bottleneck after *chenque* El Baño (Figure 6f). This is one of the most prominent narrow and deepest gorges in the area that shape mobility decisions. Then, the corridor continues through multiple paths, passing 1 km south of the La Tina rock shelter. At the Casa Piedra site (RJ136), the corridor splits into two branches. The south branch (20% of the simulations) passes through point B (Figure 6e), whereas the rest pass 2 km north. At this point, the winter corridor differs from the summer corridor, as the preferable pathways for winter are at lower elevations to avoid areas covered by snow. Next, both branches merge 3 km before the Guanaca y Luna site (point C in Figure 6e). From this last site until point D, the corridor is a narrow (0.5–1.2 km) opening at Mallín Grande (Point D, Figure 6d), where it reaches a maximum width of 4 km. For the last segment, the main branch of the corridor (96 of 100 routes) follows the contour lines near the lake until reaching the end point.

Both the summer and winter scenarios share similar probable routes on both the east and west extremes. However, they differ for the segment between the Casa Piedra site and Point C (blue box in Figure 6). In this segment, the winter corridor runs at lower elevations (closer to the lake), whereas the summer corridor shortcuts a direct route through the mountains.

## 4. Discussion

### 4.1. The Occupation of the Southern Coast of General Carrera Lake

The distribution of archaeological sites in this area follows an east to the west decline similar to the one detected in analog corridors of Central West Patagonia [3,29]. This distribution emerges as a regional pattern rooted in the antiquity of earlier colonized spaces to the east and in the presence of demographic nucleic in the open steppes of Patagonia [24,72]. Site diversity and occupation intensity are also expected to change in this geographic arrangement, as more diverse and more permanent/redundant activities are expected in landscapes with a higher degree of human presence. To date, the chronology of human occupation along the southern coast of General Carrera Lake does not extend beyond the last ~3500 years, an artifact of the few research efforts undertaken in one of the less intensely studied areas of the region [27]. However, this time span fits appropriately with the chronology of neighboring basins (e.g., Cisnes, Jeinemeni, Ibáñez, and Chacabuco River valleys, see Figure 1a) that show increased human occupation during the late Holocene period [43,73,74]. Sites and remains of scattered activities are more frequent to the east and show much more diversity, including not only residential and subsistence activities but also funerary features. To the west, the archaeological record consistently diminishes in frequency and diversity, with a marked dominance of rock art at several locations in the "Paso Las Llaves" area.

Therefore, the archaeological record presented herein, despite its newness, provides an appropriate representation of the human trajectories in the area. Its chronology constrains the occupation of the space and the mobility routes defined herein. Moreover, the 3050 to 1000 cal BP time span for the occupation of the La Tina rock shelter, a key site in the

distribution of the archaeological record and a key location in the mobility corridor through the southern coast of General Carrera Lake, suggests that this route was commonly used by late Holocene hunter-gatherers.

*4.2. Least-Cost Paths and Uncertainties*

To derive the mobility corridors of hunter-gatherers in the late Holocene, we proposed a methodology with two distinct elements. First, we explored whether seasons could affect mobility corridors. For this, we created summer and winter scenarios that included snow above an elevation threshold. Second, we explored the likely routes within the LCP model uncertainties. LCP models that derive a single LCP could be biased toward one particular assumption. Therefore, we proposed a methodology to add variability, allowing us to model a set of equiprobable routes that are within the confidence interval of data and model errors, avoiding model overfitting.

We consider this method a step forward in the study of LCP models and incorporating their uncertainties. Nonetheless, it is noteworthy that the technique does not fully explore all possibilities, and further improvements can be applied. For example, a new procedure could be to build the cost surface criterion based on probability functions or ranges for each land use category rather than on a single value. However, the dimension of the problem increases when this procedure is introduced, and more simulations are needed to explore plausible routes (increasing computational costs). Considering the assumptions and unknowns of modeling past LCPs, the approach presented in this study allows for examining a range of plausible routes. The methodology is straightforward, adding variability to the land use classes and accounting for uncertainty in the digital elevation models. The results identify mobility corridor bottlenecks, paths that, in addition to the unknowns, are more likely to correspond to historical travel routes. In contrast, the method identified areas where the mobility corridor spreads, suggesting that the path is unconstrained and/or travelers could have different semi-parallel ways.

Establishing the importance of incorporating uncertainties is necessary to note that the deterministic LCP models share the majority of the route with the most equiprobable paths from the stochastics models. Moreover, the correlation between archaeological sites and deterministic paths, specifically for the control and winter scenarios, shows us the quality of these route predictions. Hence, the methodology developed in this research allows us to understand the movement of hunter-gatherers not only through archaeological sites but also with the energy LCPs.

*4.3. Cluster and the Least Accumulative Cost Distance*

The least accumulative cost distance map is a novel approach to incorporate into the definition of cluster, in addition to the concentration of archaeological sites and the maximum distance between them (cluster methods). For example, the east-to-west movements along the southern coast of General Carrera Lake add a cost for this displacement not as a Euclidean distance but instead by considering slope and land-use cover as potential constraints to movement. In this study, we were able to identify three clusters (Figure 7) associated with the archaeological site distribution and the least accumulative cost distance map between the initial point in Chile Chico town and the endpoint close to Puerto Guadal town. The eastern cluster has the greatest concentration of archaeological sites, including residential locations, concentrated funerary features or *chenques*, hunting and processing areas, and other minor surface scatters of lithic material suggestive of open-air activities conducted across the landscape (cluster A in Figure 7). The second cluster is conditioned by the conspicuous presence of the rocky outcrop that defines "Paso las Llaves", the most prominent constraint in movement. Archaeological sites in this area are dominated by rock art paintings, paramount among them the La Tina rock shelter. Residential sites or records of other activities are absent in this area, highlighting its role as a sector of passage or internodal space instead of a place suitable for more extended stays (cluster B in Figure 7). The third cluster provided better dwelling conditions, such as less steep terrain, freshwater

sources, and a different set of resources, such as wood. The presence of several *chenques* and one surface scatter of lithic material indicate that this place was effectively occupied, despite its western position (cluster C in Figure 7). In addition, a critical factor in the definition of clusters was the presence of barriers such as the rocky "Paso Las Llaves" and deep canyon rivers such as Avilés, Los Maitenes, Las Horquetas, and Las Dunas. These barriers quickly increased the cumulative value of the cost distance, producing natural clusters.

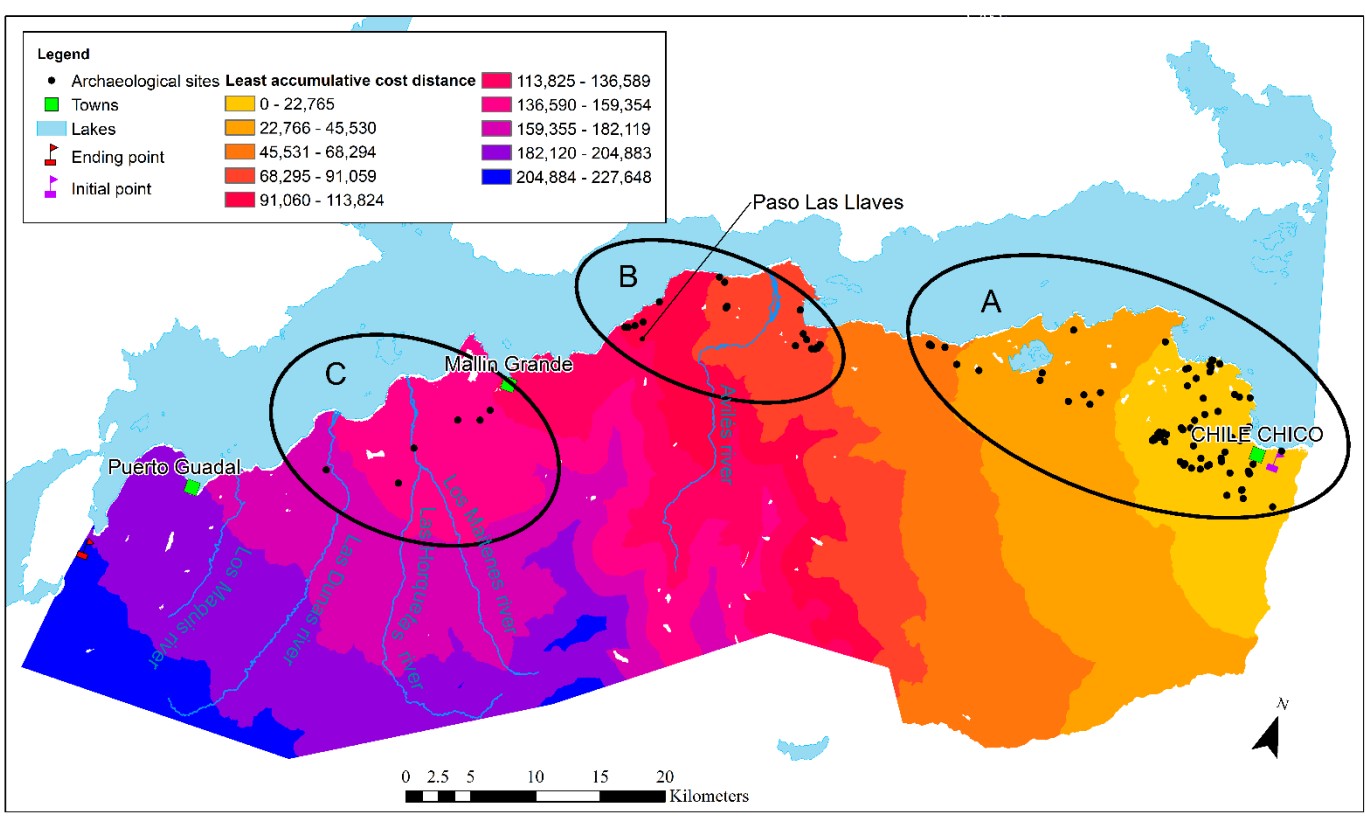

**Figure 7.** Identification of clusters (black circles) along the southern coast of General Carrera Lake following cluster and least accumulative cost–distance approaches.

### 4.4. Critical Zones for Movement along the Southern Coast of General Carrera Lake

In addition to defining clusters, several critical zones for movement in the area were identified in our study. The incorporation of the uncertainties produced alternative routes to the original LCP corridor. However, some critical places did not present passing options, thus generating a bottleneck in the model (Figure 6). These may be considered geographic barriers at a local scale. Such barriers are not unsurmountable spaces but rather narrow passes or conduits that provided people with safe passage through rougher terrain. The most critical zone is the "Paso Las Llaves", a critical point for (vehicle) movement even today. Here, the lakeshore remains the only option for crossing. The archeological record concentrates at both ends of this zone, showing the spatial correlation where the findings are indicative of no alternative path options, signaling the ruggedness of the terrain.

The criticality of "Paso Las Llaves" was exposed in Scenario 2 (the summer path), where the model prefers to move toward the highlands to avoid this rocky steep place. Future surveys will be key to assessing the use of such routes and exploring the potential of high terrain at this point. Indeed, other highland areas south of General Carrera Lake, such as the Jeinemeni Plateau, have produced lithic material, although they occur only in small surface clusters. The decision to go up by the Avilés River to the highlands is associated with a cluster of archeological events as "Casa de Piedra" or "La Tina rock shelter" (Figure 8), showing these sites as mandatory stops or, at least, key points along the route.

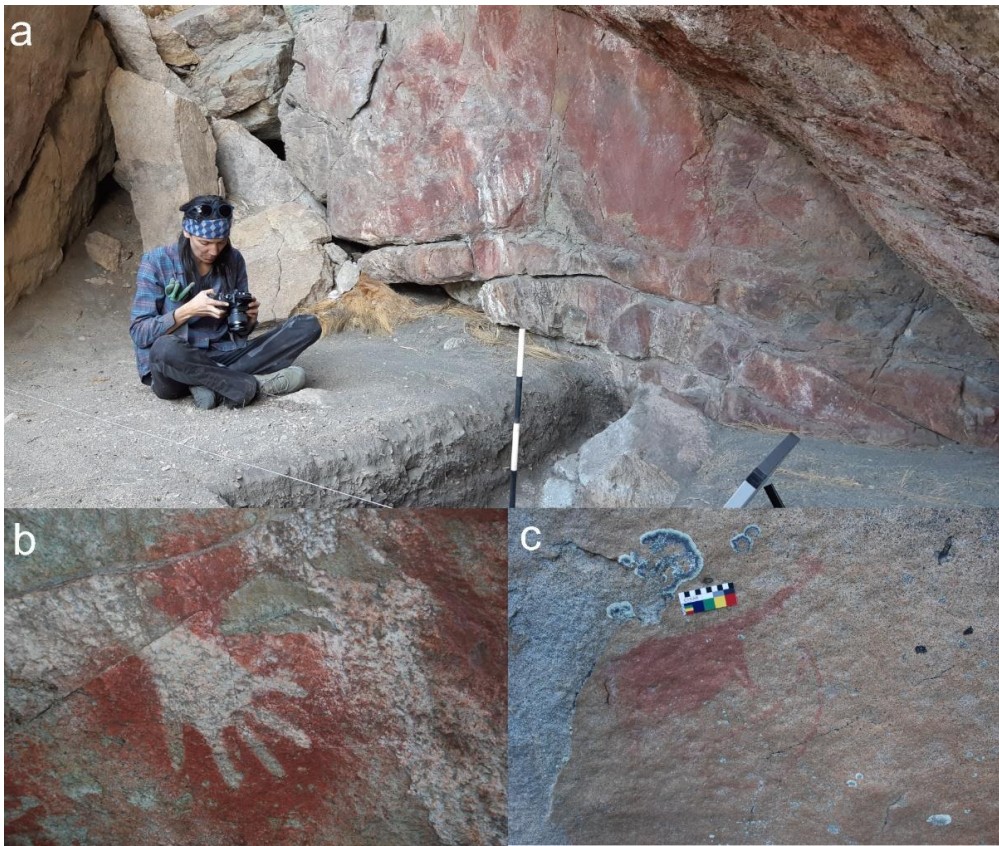

**Figure 8.** Rock art from the southern coast of General Carrera Lake: (**a**) excavations beside panel with painted hands at La Tina rock shelter; (**b**) stencil hands at La Tina rock shelter; (**c**) *Lama guanicoe* motif at Guanaca y Luna site at "Paso Las Llaves" sector.

## 5. Conclusions

The movement between occupied locations occurs along routes that connect individuals with each other, their resources and activities, thus resulting in a complex network linked to the landscape [4,75]. Although critical for understanding the use of marginal spaces, such as our study area and the rest of West Patagonia, the study of past human movement through natural corridors has remained mostly untested and has mainly been addressed intuitively [3]. This research was a significant effort to understand the movement of hunter-gatherers during the late Holocene along the southern margin of General Carrera Lake, from east to the west (and backward), not only through the identification of archaeological findings but also with the aid of LCP models. In this sense, LCP models are considered as a stage in the understanding of past mobility, which we have also address with other methods such as exotic (obsidian) and local raw material provenance and settlement patterns [26,62,63]. Relating the archaeological record and LCP showed a high spatial correlation, allowing us to infer the movements and occupation of those places with a high degree of certainty. The results presented in this paper indicate the repeated use of internodal spaces that were effectively incorporated into mobility corridors at least during the last three millennia. Redundant, albeit discontinuous occupation, is a feature detected in different sites and basins of Central West Patagonia [25,74]. As such, rock art production along the General Carrera Lake corridor, in sites such as La Tina rock shelter, shows that people were repeatedly drawn to it as they acknowledged this corridor was a conduit that allowed moving from east to west and backwards.

Additionally, the method used herein also provides indications into which spaces and routes were not originally considered for research and yet are likely used routes. Further archaeological studies of new spaces, such as surveying the summer route associated with the Avilés River, will be crucial to develop a more comprehensive idea regarding

seasonal transit over the highlands. Last, we suggest that the methodology designed for this study, which accounts for multiple solution paths, is useful to explore mobility corridor bottlenecks and the use of less constrained areas, such as flat surfaces.

**Supplementary Materials:** The following are available online at https://www.mdpi.com/article/10.339 0/land11081351/s1, Table S1: Detailed archaeological record of the southern coast of General Carrera Lake.

**Author Contributions:** Conceptualization, P.M.-M., C.M. and A.N.-D.; methodology, P.M.-M., C.M. and A.N.-D.; validation, P.M.-M., C.M. and A.N.-D.; formal analysis, P.M.-M., C.M. and I.I.; investigation, C.M. and A.N.-D.; resources, C.M. and A.N.-D.; data curation, P.M.-M., C.M. and A.N.-D.; writing—original draft preparation, P.M.-M., C.M., I.I. and A.N.-D.; writing—review and editing, P.M.-M., C.M., I.I. and A.N.-D.; visualization, P.M-M., C.M. and I.I.; funding acquisition, C.M. and A.N.-D. All authors have read and agreed to the published version of the manuscript.

**Funding:** This research was funded by ANID FONDECYT 1180306, ANID FONDECYT 1210042, and ANID Regional R20F0002.

**Data Availability Statement:** All archaeological specimens in this study are curated in the Archaeology and Heritage Section at Centro de Investigación en Ecosistemas de la Patagonia (José de Moraleda 16, Coyhaique, Chile) or at the Museo Regional de Aysén (km 3 camino a Coyhaique Alto, Coyhaique, Chile). They are available upon reasonable request with prior consultation with the authors and the collection curator of the museum.

**Acknowledgments:** We acknowledge the permits for our work and the help of Cristian Saldia, Mauricio Quercia, Sergio Haro, Sergio Giménez, Carlos Maglio, Ceferino Márquez, Ignacio Márquez, Washington Fica, Omar Fica, Vicente Sandoval, César Inallao, and Parque Nacional Patagonia (CONAF). The following colleagues participated in fieldwork: Omar Reyes, Carolina Belmar, Juan Bautista Belardi, Javier Carranza, Nicolás Araneda, Pedro Fuentes, Joaquín Crisóstomo, Sebastían Grasset, Constanza Neira, Antonia Fuenzalida, Francisca Moya, Rosario Cordero, Diego Galleani, Juan Pablo Virgilito, Gustavo Fredes, María Paz Quercia, and Matías Plaza.

**Conflicts of Interest:** The authors declare no conflict of interest. The funders had no role in the design of the study; in the collection, analyses, or interpretation of data; in the writing of the manuscript; or in the decision to publish the results.

## Appendix A. Least-Cost Path Model Sensitivity Analysis

This section evaluates the sensitivity of adding a Gaussian random field (GRF) to the cost surface. Two variables of the GRF are evaluated: integral scales (isotopically) and variance. The variance corresponds to the magnitude of the noise. If the GRF variance is too small, it will not affect the LCP model. In contrast, if the GRF magnitude is too large, the cost LCP resembles a random path. Therefore, we attempt to find a balance regarding the magnitude of the GRF. The integral scales correspond to the distance from which the Gaussian random field correlates in space. We selected the control cost surface for the sensitivity test, which is based on the slope. We run an ensemble of simulations permuting the variances var $\in$ (0.1, 0.5, 1, 2, 4, 10) degrees, and the integral scales are $\in$ (1, 5). Plots a and c of Figure A1 present two different GRFs with integral scales of 5 km and 1 km, respectively. The results from the sensitivity test showed that variances values of 1 and 2 present a compromise on the magnitude. In addition, the integral scales did not play a determinant role in comparison with the variances. Consequently, we chose a variance of 1 and integral scales of 1 km.

Figure A1 shows the GRF and the cost surfaces with the GRF. The cost surfaces (Figure A1b,d) do not offer a significant visual difference. The lack of distinction is due to the noise term, which has a Gaussian distribution (mean 5 and variance 1) and is smaller than the influence of the slope, which ranges from 0 to 90. However, the LCP algorithm is sensitive to small perturbations. For example, the LCP will be optimized by preferring a cell with a smaller cost even if it is not significant and can divert the route in the following cells. In these cases, introducing a noise term can severely affect the outcome, as seen in flat areas. Last, it is worth noting that the cost surfaces of Scenarios 2 and 3 have different

units than Scenario 1. Therefore, we scaled the variance to match the distribution of the cost surface from Scenario 1. Specifically, the Scenario 1 cost surface ranges from 0–90, and Scenario 2 ranges from 0–10; therefore, the variance was scaled accordingly to match the maximum and minimum differences for each scenario.

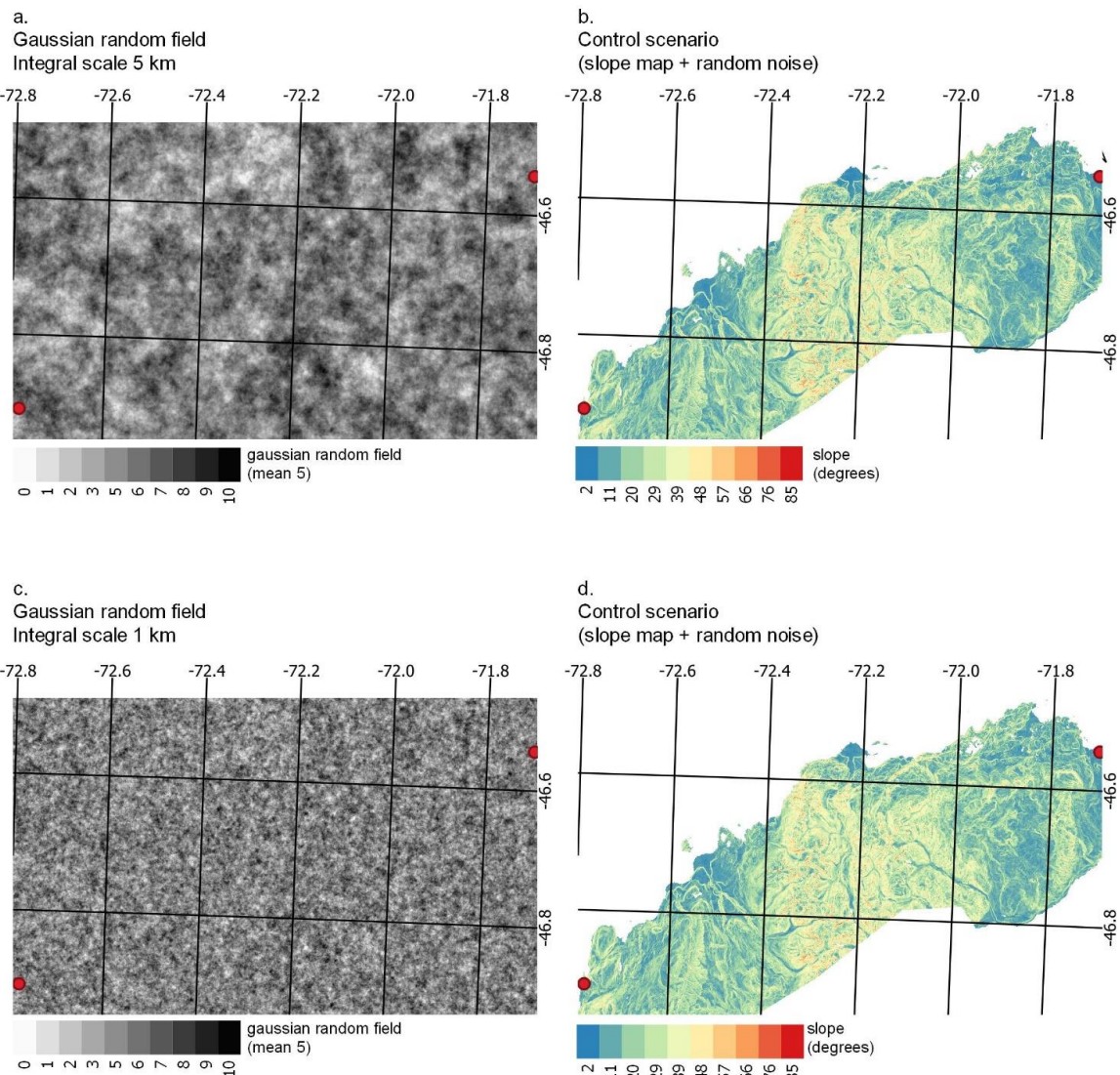

**Figure A1.** Sensitivity test for different Gaussian random fields. (**a**) Gaussian random field using an integral scale of 5 km, variance of 2, and mean of 5; (**b**) Cost surface corresponding to the addition of slope and the Gaussian random field from a; (**c**) Gaussian random field using integral scale of 1 km, variance of 2 and mean 5; (**d**) Cost surface corresponding to the addition of slope and the Gaussian random field from **c**.

To run the LCP model using the ArcGIS toolbox, we concatenated the following tools in the model builder (Figure A2). In addition, we provide a screen shot of the Python code from the model builder (Figure A3).

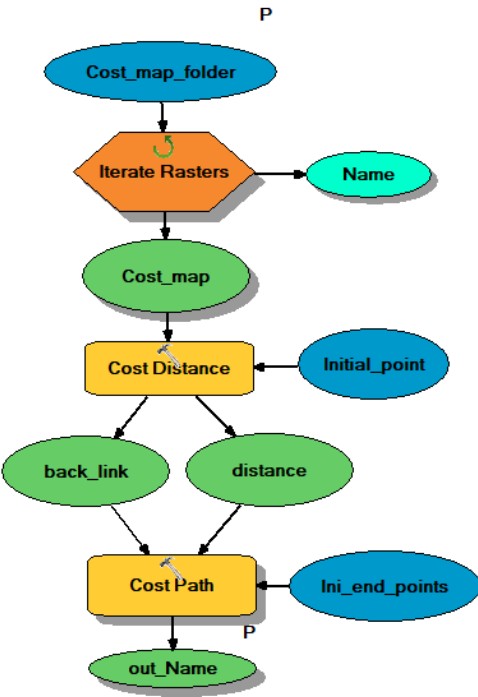

**Figure A2.** Flow diagram from the ArcGIS toolbox Model builder.

```python
# Import arcpy module
import arcpy

# Load required toolboxes
arcpy.ImportToolbox("Model Functions")

# Script arguments
out_Name = arcpy.GetParameterAsText(0)
if out_Name == '#' or not out_Name:
    out_Name = "PATH\\out_grf\\out_%Name%.tif" # provide a default value if unspecified

Cost_map_folder = arcpy.GetParameterAsText(1)
if Cost_map_folder == '#' or not Cost_map_folder:
    Cost_map_folder = "PATH\\scenario_2" # provide a default value if unspecified

# Local variables:
Ini_end_points = "points\\ini_end_points"
Initial_point = "points\\ini"
Cost_map = "PATH\\scenario_2\\Scenario_2_Sim_100.tif"
distance = "PATH\\d_grf\\distance_%Name%.tif"
back_link = "PATH\\bl_grf\\back_link_%Name%.tif"
Name = "Scenario_2_Sim_100.tif"

# Process: Iterate Rasters
arcpy.IterateRasters_mb(Cost_map_folder, "", "", "NOT_RECURSIVE")

# Process: Cost Distance
arcpy.gp.CostDistance_sa(Initial_point, Cost_map, distance, "", back_link, "", "", "", "", "")

# Process: Cost Path
arcpy.gp.CostPath_sa(Ini_end_points, distance, back_link, out_Name, "EACH_CELL", "Id", "INPUT_RANGE")
```

**Figure A3.** Python code from the model builder.

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
