# Peer review of "Past Human Mobility Corridors and Least-Cost Path Models South of General Carrera Lake, Central West Patagonia (46° S, South America)"

_land, doi:10.3390/land11081351_

Round 1
Reviewer 1 Report
The paper is interesting. Its organization with sections (Introduction, Materials and Methods, Results, Discussion, Conclusions) is adequate, and the material is ordered in a way that is logical, clear, and easy to follow. The authors cited sources adequately and appropriately, and all the citations in the text are listed in the References section.
Research data are presented and visualized in a clear way, which makes the paper readable. The methods are described in a proper way. The paper is well structured and well written.
Author Response
Reviewer 1 considered this paper is interesting, well written and organized, and with material presented in a logical, clear, and easy to follow fashion. References cited are adequate for this reviewer. Also, methods and research data are considered to be presented and visualized in a clear way, which makes the paper readable. We have nothing to add but thank this reviewer for his/her thoughtful reading and observations.
Reviewer 2 Report
I liked this paper. I thought it was a very interesting application of LCP to understand complex phenomenon regarding mobility among mostly hunter-gatherer groups, the archaeological correlates of which are ephemeral at best. I am not well-read on similar literature from this area or region, so some of my comments might be related to my overall lack of familiarity with that body of literature.
There are a few instances where I think there may have been some words or phrases that got lost in translation. I have provided detailed edits in a PDF that I hope the authors will address. Section 2.2 in particular needs attention to the use of tenses – just stay consistent.
I have a few hang-ups with the article as it stands now, which I will just briefly describe because I also address them directly in the draft.
First, there is little to no discussion of the number of sites involved in this study, nor their specific characteristics. You need to describe the MATERIALS used in your Materials and Methods section, and this includes both the digital as well as archaeological materials. As I mention in my comments, this seems a large oversight, as these sites comprise the body of the research. I would like to see a concerted description of the known sites in the area -- how many sites were there? What time periods do they date to? Where were they located in regard to one another, but also in relation to the type of landform/ecosystems they were found in? What was the degree of preservation of each site? These are factors that seem to me to be immensely important not just for the validity of your study, but also to consider for your LCP modeling. A more concerted mention of the digital materials and datasets used is also needed. Mention is made sporadically, but I think it’s better to clearly describe before you dive into the methodology.
Another item I’d like to point out -- and I suggest you pepper throughout the paper -- is a discussion of how this work is heuristic at best. LCP analysis of past behavior can never be taken as fact and is more about helping us (archaeologists) to better understand possible modes of movement. And they always need to be ground-truthed or tested in some way. Which leads me to another thought – will you be ground-truthing the results of your Monte Carlo simulated paths?
The last big lack for me is inclusion of any references to ethnographic or ethnohistorical evidence of movement in this area, or in the region. As you aptly note, LCP is based necessarily on a set of assumptions that we (today) make about past landscapes and decision-making. One way we can offset these biases is to look to the behavioral processes of sub-recent groups living and moving around similar landscapes. Relatedly, I feel that the introduction/background does not really address the issue of hunter-gatherer seasonality, although two of your main scenarios address seasonal movement. It would make this paper stronger, in my opinion, if you can draw in some literature on seasonal movement as well as territory size so that we can get a sense of where and how this study adds insight. Could it be that the area you are focusing on only represents a portion of a larger territory and was only utilized during specific seasons? Some discussion of the drivers of movement would also be good to include. Why were people moving west-to-east? Was it because of changes in resource distribution, or population growth, or something else? Where I work, there is a clear pattern of settlement favoring wetlands during some periods, and high-and-dry areas in other periods. This seems to be related to the types of subsistence being pursued. I know this paper is focused on a method to better understand mobility, but we cannot understand mobility without also understanding settlement and subsistence patterns as well.
My last global comment is about the neighborhood analysis in the Discussion. Neighborhood analysis has been used commonly in Mesoamerica and the American Southwest to understand how hierarchy/power structures movement and settlement among sedentary, complex societies. And in contexts with extremely tight C14 sequences that can clearly establish contemporaneity. I would refrain from applying this here because you are working with a dataset spanning ~2500 years and therefore cannot make assumptions about coterminous settlements (unless I am somehow reading this section wrong). What would be more interesting is to talk about why you see this clustering of occupations over the longue durée – what was it about these locations that drew people to them over and over? And furthermore, the spaces in between with very few occupations – why is this pattern as it is? Does it reflect actual settlement patterns, or is it the result of differential preservation, or research bias? This to me seems a more constructive question to ask of your data.
There are many strengths to this paper – I love to see sensitivity analyses being applied to archaeological questions – and I hope to read more from these authors in future.

Author Response
Reviewer 2 indicated he/she liked the paper, stressing that it was a very interesting application of LCP to understand mobility in our archaeological context. All observations and concerns are constructive and our careful attention to each of them was instrumental in modifying the paper to what we believe, is an improved version. Besides the general concerns, the reviewer provided detailed edits in a PDF which were entirely accepted. These details stressed in some of the general observations, suggested clarification in some points, and improved the English language. These were carefully followed and were highly beneficial for our paper.
Major points
- “There is little to no discussion of the number of sites involved in this study, nor their specific characteristics”
We have now included: a revised 2.2. Archaeological Material and Methods section with a concerted description of the known sites in the area (and environs). These includes numbers of sites, percentages, time ranges, observations on type of sites and materials recorded. Some of these data in also summarized in a new table inserted in the manuscript (see Table 1). Also, we have provided detailed information of the assemblage of sites considered in the analysis by including a new Table S1 (as supplementary online material). The precise location and altitude are included for each entry. New references were included accounting for the original recovery of some of that material as well. The new 2.2. section also includes a complete segment that was suggested to be moved from the discussion section into the methods section. Additional observations on methods, collection criteria, and taphonomy were also included.
- “A more concerted mention of the digital materials and datasets used is also needed”. As suggested by the reviewer, section 2.3. (and all subsections therein) were written again to clarify readership and to address the different stages in the methodology and the products resulting from it. We think this new arrangement provides a much more detailed and ordered presentation of the steps considered in the analysis.
- “how this work is heuristic at best. LCP analysis of past behavior can never be taken as fact and is more about helping us (archaeologists) to better understand possible modes of movement”. Yes, we agree with the reviewer and have included comments in this line in the introduction and conclusion sections.
- “Will you be ground-truthing the results of your Monte Carlo simulated paths?
We worked with a deterministic and stochastic LCP analysis. We incorporated uncertainties, such as noise, into the cost mobility criterion to produce multiple possible mobility routes in our study area. In the end, those routes are the results of models. Models generally represent reality, as such models are not the truth but allow us to find explanations for certain patterns. The results of these models may be tested with different methods: Monte Carlo simulations, a non-parametric statistic method, is one of the possibilities to check the results.
- “The last big lack for me is inclusion of any references to ethnographic or ethnohistorical evidence of movement in this area”. There are no available ethnographic or ethnohistorical accounts for the study area. However, we have resorted on information at the macro-regional scale for such purpose. A new paragraph in the introduction deals with this issue and includes several new references that address the questions raised. This particularly draws on the overall lack of historically known groups in the area, which is a feature of mountainous areas of western Patagonia: the avoidance of this spaces. However, as we stated in the article and the other works we refer to, the archaeological record is present in a form of a geographical marginal occupation. In that line, the reviewer correctly suggested that the issue of seasonality was key. The introduction now addresses seasonality by citing the available work on that topic. The area we focus on only represents a portion of a larger territory and was likely utilized during specific seasons. This is now more clearly stated at different parts of the article because the new inclusions articulate better with the original (more synthetic) version. Also, we have been more explicit on the drivers of movement into marginal spaces y quoting population growth within one of the sentences in the introduction.
- “My last global comment is about the neighborhood analysis (…) I would refrain from applying this here because you are working with a dataset spanning ~2500 years”. We acknowledge the oversight of this problem and as the reviewer indicates, the concept of cluster is the appropriate to use in this case. The concept of “neighborhood” in archaeology refers to a different situation and context and was inappropriately used in our first version. We decided to erase it completely from our article and now it is more consistent.
- “What would be more interesting is to talk about why you see this clustering of occupations over the longue durée – what was it about these locations that drew people to them over and over? And furthermore, the spaces in between with very few occupations – why is this pattern as it is?” We have included additional comments (and citations) in the conclusion section to address this point. We think that this observation has been very useful in improving the overall interpretation of the patterns presented in this article.
Some specific comments:
- As for the reference of Herzog's work on LCP analysis. We thank the reviewer for this suggestion. Herzog's work is interesting with several methodological steps. We have improved our manuscript with some key points from it. Both references suggested were included.
- All changes suggested to Figure 1 were considered (sub figure d box, elevation of precipitation, basin outline).
- The least-cost path and LCP (acronym) were reviewed and consistently applied across the paper. We used the term least-cost path in titles and LCP within the paragraphs.
- As suggested, sections 2.2. and 2.3. (and all subsections therein) were written again to clarify readership and to address the different stages in the methodology (archaeological and digital) and the results deriving from it.
All other minor points were changed as it clearly shows in the new version of the manuscript provided with the track changes function shown.
List of specific comments (lines numbers refer to the original manuscript submitted) from the PDF provided by reviewer 2:
Lines 34-35: “I think your theoretical development would benefit from the reading of Robert Whallon's excellent book "Information and its Role in Hunter-Gatherer Bands" (2011)” We have included the reference to two of the works in this book. Both are placed in the introduction. Our article has benefited greatly with this suggestion. One of these papers, is Scheinshon´s chapter on rock art and movement from an area in Patagonia further north from our study area. Thank you.
Line 45: for the localization of the archaeological record. I do not understand what you mean here. We changed the sentence for clarity.
Line 51: I would also reference some of Irmela Herzog's work on issues with LCP analysis. Herzog's work is meaningful in outlining methodological steps. We improved our manuscript with some key points of Herzog's work. Both references included.
Line 52: do you mean successful? We made changes for clarity indicating what was the prediction we referred to in this citation.
Line 55: as well as the quality of our datasets, which were often collected for reasons very different than the purposes we put them to. This is correct. However, the emphasis in this sentence is related to the use of current information to predict past conditions. We made changes for clarity.
Line 57: Another source of uncertainty comes from digital elevation models resolution and uncertainty. The paragraph was modified.
Line 61. and what about scale? The observation was incorporated.
Line 66. I'll mention this again, but you might want to look at circuit theory as a corollary method for investigating multiple routes of movement. Developed by ecologists for understanding species movement through habitats but has been applied to archaeological contexts in interesting ways. see Howey 2011 JAS. As for the consideration of circuit theory and its application to archaeology, we decided to include it in the introduction, alongside Howey’s 2011 reference.
Line 85 Constraints always sounds negative to me, like ancient peoples had some kind of aversion to the place. But in reality, these may have been conduits or places of flow that represented possibilities rather than restrictions. The concept of constraints was attenuated throughout the paper as suggested by the reviewer, because of its potential negative meaning (see different sections).
Line 93 In Figure 1, you use the unit 3,968 mm which is quite a bit less than 6000 mm. Could you change the label to say ~4000 mm, just to be more accurate, or perhaps >4000 mm? We made the correction in the text and in figure 1 for consistency.
Line 108. Figure 1. this box really had me confused because in the "d." call-out, it is not oriented cardinally. Can you rotate this box so that it is oriented to the correct angle that we see displayed in part d.? We made the correction.
Line 108. Figure 1. shouldn't the basins be yellow? It is confusing because the mountain tops are also white and I'm not sure where you're referring to. We made the correction.
Line 108. Figure 1. I don't know this term. With respect to the “Packraft”, this is a colloquial term for a small, portable, inflatable boat designed for use in all bodies of water. A packraft is designed to be light enough to be carried in a backpack for extended distances. We clarified this in the text.
Line 114. define "pack raft". Also, on Figure 1, this term is spelled "packraft" -- be consistent. The term was defined in the manuscript.
Line 134. most everywhere else, you refer to LCP as "Least-cost path" -- pick one version and stay consistent. We changed the tittle and the uses of the LCP acronym throughout the paper.
Lines 135 to 140. The further I read into the next section (2.3.1 and 2.3.2), the more confused I am by this short paragraph. It doesn't seem that what you are introducing here with your "first" and "second" clearly correspond to what we read in 2.3.1 (what I assumed with be the first part) and 2.3.2. We wrote the section 2.3 again to provide better readership (see general comments above).
Line 135. in the previous section, you write in the past tense. Here, you write in the present tense. I would pick one tense and carry forward consistently in the rest of this section (2.2). Modified L135-140.
Line 138. Do you mean that you explored routes that fall within particularly problematic areas for LCP? Modified L135-140.
Line 140. to my understanding, equi-probable would mean equally probable, based on the concept of equifinality. Could you state this more clearly? That you modeled a number of potential routes and that some may come up looking the same even though different weighting schemes underpinned them? And also, why is this important to include in your study? The paragraph (L 135-140) has been modified. In addition, the importance and why we are pursuing this methodology have also been expanded in the introduction of the article.
Line 141. I would refer to this as a cost surface rather than a map. A map to me is something more finished and the cost surface is something we generate as an input for our LCP models. I might also rename this section something like "Developing Cost…” We changed the concept to “cost surface” as was required. This is shown throughout the text and in figure 2 for consistency.
Line 142. stay consistent with how you refer to LCP when you spell it out
We used the term Least-cost path in titles and LCP in the paragraphs.
Line 143. I'd say the first key step
We changed it in the manuscript
Line 143. again, I think cost surface is the better term
We changed to cost surface as was required.
Line 143. awkward wording. I might say "degree of human effort (represented through various measures, such as time or calories) involved in moving across the landscape..."
Thank you, we have included this correction. The suggestion from the reviewer has more details.
Line 151. what do you mean by this?
Extra Andean steppe is a geographic location east of our study area. We have clarified this concept throughout the paper.
Line 152. I would just note here that the movement you are describing is not west-to-east but rather southwest to northeast.
The movement is northeast-to-southwest but we used in a general point of view east-to-west as other archaeological studies in Central West Patagonia. However, we have pointe dout the northeast-to-southwest distribution of the corridor in this case.
Line 153. I'm just wondering how these match up with possible models of seasonal movement. Can you talk a bit more in the introduction/background about what parts of this region would have been used in different seasons? Also, is it possible, looking at the archaeological record, that this area was only used during one part of the year? I'd imagine the lake is full of fish and thus perhaps people focused on this area during times when fish are most abundant. I don't think you ever address this point directly, and there seems to be an unstated assumption that this area somehow represents a total habitat range for some hunter-gatherer group, when we know that movement likely took place at multiple scales and rates, hence your corridors. Maybe talk more about where people might have been going with these corridors.
The introduction now addresses seasonality by citing the available work on that topic. The area we focus however, on only represents a portion of a larger territory and was likely utilized during specific seasons. This is now more clearly stated at different parts of the article. Regarding lakes, in this region they were not systematically used by indigenous populations, either for navigation or for resource consumption. Both archaeofaunal records and isotopic data clearly stress the selection and consumption of land resources. Also, there is no indication of the required technology for the appropriation of water resources in the assemblages studied across this microregion, nor communication with coastal peoples at this latitude.
Line 154. Uses
Change was made.
Line 154. Only
Change was made.
Line 159. do you also use slope and land use here? Be explicit
Change was made.
Line 167. Class
Change was made.
Line 168. Did you look at any ethnographic data to support your cost designations? Or ethnohistorical data? This to me would add greater support to your classifications which for now are just based on modern-day ideas of movement...
Given the lack of ethnographic, data we added more supporting modern information to the manuscript. In summary, we created a relative ranking of human effort for the land use class following the current movement in the zone and assimilating the relative ranking of others similar studies.
Line170. here I would be clearer about using the weighted overlay tool in order to merge the to input surfaces and generate a composite effort surface
We did not use a specific weighted overlay tool just raster algebra, as is explained in the manuscript.
Line 171. What does this mean? Units of what?
Because is a relative ranking, it is dimensionless. Based on Pandolf et al. (1977) cited in Herzog, I. (2014) the energy expenditure for walking in 35cm of snow is four times as high as on a tarred road.
Line 174. class
Change was made.
Line 174. I would make a note somewhere that you had to make assumptions about what the original land cover of present-day agricultural and urban areas were
We wrote the inclusion about inexistent coverage in the present-day but the today land use map is our best approximation.
Line 180. are prone to?
We have modified the sentence.
Line 180. I would say this is less of an issue than the biases we (the LCP modelers) introduce -- the model is only as good as the data we give it
We agree that main issues come from the definition of the cost surface. The sentence has been modified.
Line 180. I like this term, can you use it in the above section once or twice to refer to your input criteria?
Yes. We have incorporated the term “cost criterion” on the introduction and methodology.
Line 181. I'm confused by this -- these biases ARE incorporated into our models, just look at your Table 1. I'm not shirking it, we all have to make these judgment calls, but they do impact the overall model output. Maybe I am not understanding what you mean here.
The sentence has been modified. Please refer to previous comment.
To clarify. In practice we are “perturbing” or “adding a noise term” to the cost surface for each iteration. The added variability represents the “aggregated uncertainties” of the cost surface. For example, the slope of each pixel could have an error. In addition, the cost assigned to each class is homogeneous, then the noise term adds variability for each pixel between each class. Yet, as mentioned earlier, if following this procedure is key to perform multiple realizations.
Lines 185 to 200. I am not familiar with this procedure, but it sounds to me like you might want to look into Circuit Modeling, which also provides a variety of possibilities for movement based on the overall conductivity of the cost surface. See for example MCL Howey's 2011 article in Journal of Arch Science and McLean and Rubio-Campillo's 2022 article in the same journal.
We thank the reviewer for pointing out about circuit theory. Indeed, the procedure followed in the manuscript outputs a set of likely routes, and in this sense, it resembles to circuit theory.
We have added in the Introduction section a description of the pros/cons of circuit theory and LCP. In addition, we have modified the methodology section to better explain the procedures.
Is important to notice that both approaches LCP and Circuit Theory (CT) have limitations.
Citing (McLean & Rubio-Campillo, 2022): “It is important to understand what CT cannot show. The current values represent potential mobility, but an area of high current in the output does not indicate that the area was well connected in reality, as this potential mobility may not have been utilized.”
Therefore, one of the main limitations of Circuit Theory is that it does not provide “a route” but rather areas of high mobility potential that may have not been used as passage areas. Here we chose to simulate multiple LCP, a methodology similar to Lewis et al 2021 as we consider relevant to identify a likely route rather than likely potential mobility areas which could be isolated.
Is important to note that we do not claim (please refer to Discussion section) that our methodology solves all issues of LCP. We consider fundamental to simulate multiple LCP providing the reader a better (fair) understanding of areas with low/dense likelihood of passage.
Line 185. These are ways you are trying to alleviate uncertainty, right? Incorporate sounds like you are trying to include them but I think the opposite is the case.
Yes, we have changed the sentence.
Line 188. often?
Yes, sentence has been modified.
Line 193. is this a word?
It is not! Changed to “increasing”
Line 215. again, circuit theory does something very similar, requires less processing
As correctly pointed out by the reviewer, calculating multiple LCP is computationally expensive. However, computing times were not prohibitive. We coded and automatized to run all LCP calculations which took between 20 - 30 minutes.
Line 218. Do you have evidence of travel paths that date back that far? Or are we talking modern-day routes? Here again it might be useful to refer to ethnographic/ethnohistorical data
We were talking in general, not a specific historical route or actual route.
Line 230. You lost me here, but it sounds like a promising strategy.
Yes, it is a strong non parametric method. In other words, the method compares the observed statistic versus several simulated statistics, creating a distribution of probability. In this case, the statistic is the average distance of sites to the LCP.
Line 318. described above in Section 2.3.2
Change was made.
Line 322. Specifically,
Change was made.
Lines 345 to 346. I was going to say... how does the lack of archaeological work and perhaps also site visibility skew this away from the uplands?
Exactly, we do note the bias. Thus, future surveys are incorporated as future steps in conclusions.
Line 350. I know you've mentioned that movement generally followed a west-to-east thrust, but weren't people (especially hunters and gatherers) likely moving back and forth? Or do you assume that moved back west through some other area?
Indeed, we think that movement was in two ways. We have now clarified this. We stress one way just to indicate that most dwelling occurred in the east and that seasonal movement to the west was a minor feature in mobility.
Line 423. do you mean to say that all LCPs had to proceed by this site? or that is was an important location?
Both, it is a common place for LCPs but at the same time has a crucial importance archeological.
Line 433. again, I think this is redundant
Change was made.
Line 440. when this procedure is introduced
It is a new procedure for future studies. We rewrote the sentence for clarity.
Line 446. real? or actual travel routes used in the past?
Historical. Change was made.
Line 451 and 453. “ic”.
Changes were made
Line 457. I would be very wary of trying to find neighborhoods among clusters of sites that might range in their use date by thousands of years. Unless you have a tight C14 sequence to indicate co-occupation of these sites, I would remove this section altogether. Or, if you want to keep it in, I would use some term other than neighborhood because that suggests you are working on sedentary, complex societies. See e.g., work by Michael E. Smith, such as his 2010 article in Journal of Anthropological Archaeology
We acknowledge the oversight of this problem and as the reviewer indicates, the concept of cluster is the appropriate to use in this case. The concept of “neighborhood” in archaeology refers to a different situation and context and was inappropriately used in our first version. We decided to erase it completely from our article and now it is more consistent.
Line 492. maybe this is annoying, but constraints sounds negative. Could we call these passes or conduits that provided people with safe passage through rougher terrain?
The concept of constraints was attenuated throughout the paper as suggested by the reviewer, because of its potential negative meaning (see different sections).
Despite some comments might have been related to an overall lack of familiarity (as the reviewer stated) with the archaeology of Patagonia, these more “general” comments were useful for addressing the article to a broader context of readers. We are thankful for that.
Round 2
Reviewer 2 Report
Thank you for addressing so clearly my concerns with this paper.